# Using Deep Neural Networks for Predicting Age and Sex in Healthy Adult Chest Radiographs

**DOI:** 10.3390/jcm10194431

**Published:** 2021-09-27

**Authors:** Chung-Yi Yang, Yi-Ju Pan, Yen Chou, Chia-Jung Yang, Ching-Chung Kao, Kuan-Chieh Huang, Jing-Shan Chang, Hung-Chieh Chen, Kuei-Hong Kuo

**Affiliations:** 1School of Medicine, College of Medicine, I-Shou University, Kaohsiung 82445, Taiwan; cyyang@ntu.edu.tw; 2Department of Medical Imaging, E-Da Hospital, Kaohsiung 82445, Taiwan; 3Department of Psychiatry, Far Eastern Memorial Hospital, New Taipei City 22060, Taiwan; panyiju0211@gmail.com; 4Institute of Public Health, School of Medicine, National Yang-Ming Chiao-Tung University, Taipei 11267, Taiwan; 5Division of Medical Image, Far Eastern Memorial Hospital, New Taipei City 22060, Taiwan; yen055065@gmail.com; 6Department of Radiology, Taitung MacKay Memorial Hospital, Taitung 95054, Taiwan; jeroyang@gmail.com; 7AI Lab, Quanta Computer Inc., Taoyuan City 33377, Taiwan; Will.Kao@quantatw.com (C.-C.K.); Mark.Huang@quantatw.com (K.-C.H.); okmegy@hotmail.com (J.-S.C.); 8School of Medicine, National Yang-Ming Chiao-Tung University, Taipei 11267, Taiwan; 9Department of Radiology, Taichung Veterans General Hospital, Taichung 40705, Taiwan

**Keywords:** age prediction, sex prediction, deep learning, chest radiograph

## Abstract

Background: The performance of chest radiography-based age and sex prediction has not been well validated. We used a deep learning model to predict the age and sex of healthy adults based on chest radiographs (CXRs). Methods: In this retrospective study, 66,643 CXRs of 47,060 healthy adults were used for model training and testing. In total, 47,060 individuals (mean age ± standard deviation, 38.7 ± 11.9 years; 22,144 males) were included. By using chronological ages as references, mean absolute error (MAE), root mean square error (RMSE), and Pearson’s correlation coefficient were used to assess the model performance. Summarized class activation maps were used to highlight the activated anatomical regions. The area under the curve (AUC) was used to examine the validity for sex prediction. Results: When model predictions were compared with the chronological ages, the MAE was 2.1 years, RMSE was 2.8 years, and Pearson’s correlation coefficient was 0.97 (*p* < 0.001). Cervical, thoracic spines, first ribs, aortic arch, heart, rib cage, and soft tissue of thorax and flank seemed to be the most crucial activated regions in the age prediction model. The sex prediction model demonstrated an AUC of >0.99. Conclusion: Deep learning can accurately estimate age and sex based on CXRs.

## 1. Introduction

Predicting age and sex using various medical imaging modalities has been implemented for decades. In forensic medicine, dental and manubrium with age correlation [1,2] and costal cartilage mineralization studies [3,4] have extensively been used to estimate age and sex. Bone age study is commonly used for clinical purposes to evaluate the skeletal maturity of children or adolescents [5]. Beyond age and sex per se, neuroimaging-derived age prediction has been found to correlate with influences from various diseases, such as cognitive impairment and schizophrenia [6,7,8,9,10,11]. T-scores obtained from bone density with DEXA scans among both sexes, which usually decrease with age, can be used to predict osteoporosis and fracture risk [12]. These prior studies suggest that various medical imaging modalities contain features that may have correlations with chronological age and sex.

The advantages for medical imaging to predict age and sex may have roots in the fact that age and sex are not simply one single concept. In fact, image findings may reflect the overall effects of chronological, biological, and/or pathological changes [13,14,15]. Chronological changes reflect natural development over time, e.g., endochondral ossification in skeletal evolution, changes in body shape, or breast growth related to hormones. Biological changes indicate cumulative effects resulting from interactions between the body and environment or lifestyle, e.g., obesity-related to an unhealthy diet and osteoporosis-related to diet and a lack of exercise. Beyond the physiological changes, including chronological and biological ones, pathological changes can be related to a certain disease, cohort, or sex-specific issue, e.g., cancer growth and chronic tuberculosis sequela usually seen in elderly persons and breast cancer mostly seen in females. Human eyes cannot distinguish complex changes in imaging results from the intertwined components of chronological, biological, and pathological changes, and machine learning (ML) techniques, by contrast, have the potential for detecting these corporate changes in a quantitative way.

Numerous advances have been implemented in conventional ML for bone age [16,17] or neuroimaging-derived age prediction [18,19,20]. Recently, deep learning, an ML technique that utilizes feature-learning methods with multiple levels of representation, has also been used for bone age [21] and brain age prediction [22] with comparable accuracy. Both methods can be used to summarize and quantify the continuous change in the images, which are beyond the capacity of unaided human eyes. Conventional ML methods require preprocessing and predefined features for training. In contrast, deep learning via convolutional neural network (CNN) can extract related features from raw images for further quantification analysis. Through multiple layers in deep learning models, complex input data can be transformed to an output classifier that determines specific representations of the data with multiple levels of abstraction [23]. Therefore, deep learning method may have advantages over conventional ML approaches and can be a robust method without prior knowledge.

Chronological age has been widely used in various studies to predict disease and treatment prognosis. However, biological age may better reflect the impacts of lifestyle, nutrition, multiple risk factors, and environmental factors, which, in turn, can benefit the prediction of disease and prognosis both theoretically and practically [24,25,26]. Among all medical imaging modalities, the chest radiograph is the most widely available modality, displaying plenty of information about the cardiopulmonary and musculoskeletal system. Despite research using chest radiograph-derived biological age estimates to successfully predict longevity, long-term mortality, and cardiovascular risk [13,14,27], very few studies have ever validated chest radiography-based age prediction methods [15,27], and none of them have validated the prediction model in a real-world clinical dataset. Therefore, there remains a lack of direct proof or predictive performance data regarding whether and how chest radiographs can provide accurate age and sex predictions based on clinical datasets. To fill this gap, we aimed to assess the accuracy of a deep learning model for age and sex estimation based on chest radiographs of a healthy adult cohort from a real-world clinical dataset.

## 2. Material and Methods

### 2.1. Ethics Statement

This study was approved by the Far Eastern Memorial Hospital Institutional Review Board (FEMH 107094-E), and the requirement to obtain informed consent was waived due to its retrospective nature. The participant records were anonymized and deidentified before analysis. This project was supported by Quanta Computer Inc. (Taoyuan, Taiwan). The data and analysis were controlled by authors independent of Quanta Computer (C.-Y. Y. and K.-H. K.). Three authors (C.-C. K., K.-C. H., and J.-S. C.) are employees of Quanta Computer but had no control of data presented in this study.

### 2.2. Image Data Acquisition

In this study, we used data of 50,575 individuals (age > 19 years) who underwent frontal chest radiograph studies (*n* = 71,878) for health screening from January 2008 to December 2017 at Far Eastern Memorial Hospital. To train the model purely based on physiological age change, an in-house deep learning screening model was used to exclude all images revealing any kind of lung diseases, acute traumatic lung, surgical conditions, or any type of implants. In total, 66,643 images from 47,060 individuals were used for model training and testing (Figure 1 and Figure 2). The Digital Imaging and Communications in Medicine (DICOM) files were downloaded from the picture archive and communication system of Far Eastern Memorial Hospital. The age and sex of the participants were extracted from the DICOM metadata. The images from the DICOM files were extracted in Portable Network Graphics image format and then randomly split into training (*n* = 53,315), validation (*n* = 6664), and test (*n* = 6664) sets. Multiple images from the same participant at different timepoints were selected only for the training set, rather than for the validation or test sets (Table 1).

### 2.3. Study Population Demographics

In total, 47,060 individuals (22,144 men (47.1%) and 24,916 women (52.9%)) were included. Image-based demographics are summarized in Table 1. Their mean chronological age at the time of imaging and standard deviation was 38.7 ± 11.9 (age range, 20–93 years). The distribution of the chronological age and sex of the training, validation, and test sets is shown in Figure 3.

### 2.4. Deep Learning Algorithm Architecture

The neural network architecture used in this study was Inception-ResNet-v2 (https://github.com/keras-team/keras/blob/master/keras/applications/inception_resnet_v2.py; accessed on 1 July 2021) [28], which is the combination of two ideas: residual connections [29] and inception architecture [28,30,31]. Residual connections are necessary for deeper networks to be successfully trained because the issue of vanishing gradients is greatly alleviated. Inception networks leverage different kernel sizes to extract features in different scales. The last fully connected layer of the original Inception-ResNet-v2 needed to be adapted for our learning tasks. In the sex prediction model, the number of output nodes was 2, whereas in the age prediction model, only one output node was needed because the learning task was a regression. The activation functions and how to train the networks are described in the following section.

### 2.5. Training of the Deep Learning Algorithm

Before being fed into the network, the images were resized to a resolution of 512 × 512 pixels. Grayscale values were normalized from (0, 255) to (−1, 1) by dividing the values by 127.5 and then subtracting 1.0. We used a CNN architecture (InceptionResNetV2) [32] with parameters initialized by pretrained weights optimized for ImageNet [33]. Features were selected to obtain the best model fit as follows. Ages are bounded values; therefore, the activation function of the output layer had to be bounded as well. The use of the activation function had an additional advantage: the underlying age constraint was inherent to the model structure. The widely used sigmoid was selected as the activation function in the age prediction model, whereas a softmax output function was used in the sex prediction model.

The codomain of sigmoid is (0, 1); therefore, the outputs were interpreted as min-max normalized ages. Specifically, x, $\tilde{x}$, a, and b denote age, normalized age, minimum age, and maximum age, respectively. The age was then normalized as
(1)$$\tilde{x}=\frac{x-a}{b-a}\;$$

If $x\in \left(a,b\right)$, then $\tilde{x}$ ∈ (0,1). Therefore, we normalized the true ages by using Equation (1) and fitted the model to the normalized ages. By inference, the outputs were denormalized as
(2)$$x=\tilde{x}\left(b-a\right)+a\;$$

The oldest and youngest ages in the dataset were normalized to 0.9 and 0.1, respectively. In our case, the youngest age was 21, whereas the oldest was 93. After these values were substituted into Equation (2), a = 12 and b = 102, i.e., the predicted age range of our model was 12–102 years.

For data augmentation, a rotation of up to 5° degrees was applied for each image in the training set before being fed into the model. The loss function of the age prediction model was the mean absolute error (MAE), and the model training goal was to minimize loss. The overall model performance was also evaluated using MAE. We used an adaptive moment estimation (ADAM) optimizer with a default parameter setting: β1 = 0.9 and β2 = 0.999. The model was trained with minibatches of a size of 32 and an initial learning rate of 0.0001. The open-source deep learning framework Tensorflow (https://www.tensorflow.org/; accessed on 1 July 2021) was used to train and evaluate the algorithms.

### 2.6. Gradient-Weighted Class Activation Mapping

We used gradient-weighted class activation mapping (Grad-CAM) to visually explain the most activated region produced by the model [34]. Although Grad-CAM was designed to solve classification problems, it could be adapted to visualize regression activation with a slight modification. Grad-CAM is a weighted combination of forward activation maps followed by a rectified linear unit (ReLU). Applying a ReLU in a classification scenario is reasonable because only the features that positively influence the targeted class are of interest. In the case of regression, however, negative features also affect the output. Therefore, we replaced the ReLU with an absolute value to generate an activation map for regression.

To visualize the activated region variance across diverse age levels, individuals were divided into separate 5-year age groups as follows: the bounding box coordinates containing the chest region of each individual were obtained using an in-house object detection model developed based on YOLOv3 [35] to detect the coordinates, width, and height of the lung field box. Average coordinates were then calculated for each corner of the box bounding the lung field from the entire test set, and the averaged corner coordinates were used as the reference lung field box in the template. Each individual was then registered to the template space by using the moving least squares deformation method [36], with the corners of the lung field bounding box as a control point set for each individual and as the deformed positions in the template. Finally, the transformed images were averaged in each age group to generate group radiograph templates and group summarized activation maps (SAM) (Figure 4).

### 2.7. Statistical Analysis

Data are presented as means and standard deviations for continuous variables and as percentages for categorical variables. The reported performance assessment of the model used age predictions on the test set. The model performance was evaluated by calculating the Pearson correlation coefficient (*r*) between chronological and predicted ages, the MAE, and root mean square error (RMSE). A Bland-Altman plot was used to distinguish paired differences of the model estimate from the chronological age over the paired means. The area under the receiver-operating characteristic curve (AUC) and standard errors were calculated to examine the validity of chest radiographs in sex prediction. The absolute errors and squared errors between different sex groups were compared using the Student’s *t*-test. A *p*-value less than 0.05 was considered significant, and Bonferroni adjustment was conducted in the case of multiple comparisons. The statistical analysis was performed using R (version 3.5.3).

## 3. Results

### 3.1. Age Prediction

Regarding the age prediction performance for the test set, when model predictions were compared with chronological ages, the mean difference, MAE, and RMSE were 0.0, 2.1, and 2.8 years, respectively. Age prediction results are shown in Figure 5, and four illustrated examples are shown in Figure 2A–D. Figure 6 demonstrates the Bland-Altman plot for the difference versus mean between the model estimates and chronological ages over the mean of the two estimates. The mean difference between model estimates and the ground truth was close to 0 (−0.04) years, with a standard deviation of 2.77. The difference was consistent for averages <60 years, but the difference decreased as the average increased beyond 60 years. Both MAE and RMSE were slightly smaller in the female group (2.1 and 2.7, respectively) than in the male group (2.2 and 2.8, respectively), with *p* values = 0.08 and 0.04, respectively.

### 3.2. Sex Prediction

The sex prediction model examined with the test set achieved an AUC of 0.9999943 (95% confidence interval = 0.9999876–1.0000000) (Figure 7).

### 3.3. Group Summarized Activation Maps

Figure 8 showed pairings of group-SAM, group-averaged template, and the corresponding superimposed activation maps by age. The figure shows the key regions relevant to the predictions highlighted. For the age prediction model, the most relevant anatomical regions were the cervical, thoracic spines, first ribs, aortic arch, heart region, rib cage, and soft tissue of the thoracic wall and flank. Regarding the sex prediction model, the most relevant regions were the breast shadow and lower neck (image not shown).

## 4. Discussion

This study developed a deep learning model for age and sex estimation based on chest radiographs from a healthy adult cohort in a hospital-based real-world clinical dataset. By using CNNs, our model predicted age and sex with high accuracy (Age: MAE/RMSE = 2.1/2.8 years; Sex: AUC = 0.9999943) and the mean of the differences between model estimates of age and the ground truth was close to 0 years. In addition, the modified Grad-CAM method with SAM demonstrated the variance across different age groups in the most activated anatomical regions, containing critical features for the age prediction model.

Under the assistance of a CNN, and using regression method, our chest radiography-based age prediction model resulted in an MAE and an RMSE of <3 years and showed a strong correlation between predicted and chronological ages. With a similar method, Larson et al. [21] developed a hand-age assessment model with an MAE of 0.5 years and RMSE of 0.63 years. Instead of chronological age, the evaluation target of the hand age study was skeletal maturity, defined by image findings, and the ground truth was also determined by human reviewers based on images. Therefore, compared to the prediction of chronological age, a model predicting skeletal maturity may be easier to comprehend because the evaluation target and ground truth are both image-based. In another brain-age prediction study [22], the models were trained using a brain magnetic resonance (MR) images dataset (*N* = 2001) with an MAE > 4 years and RMSE > 5 years. Although speculative, it seems possible that our model, demonstrating greater accuracy, may extract more relevant features for age prediction from the chest radiographs than those extracted in brain MR images.

All prior studies predicting age and sex were based on open datasets, which have been criticized for mislabeled or poor-quality images [37,38]. In those prior studies, researchers did not distinguish images of disease conditions from those of healthy individuals [15,27]; therefore, image information from pathological changes may be mixed with those from chronological and biological changes. To develop a model that minimizes the impacts of non-physiological changes, we used only high-quality chest radiographs and excluded images from people with lung diseases, trauma, surgical condition, or any type of implants in this study, which produced a relatively good accuracy (age: MAE/RMSE = 2.1/2.8 years; sex: AUC = 0.9999943) (Figure 5, Figure 6 and Figure 7). However, one should be cautious that the MAE/RMSE cannot be directly compared across different studies and the high accuracy results in the current study are not simply because of the usage of high-quality images; instead, other factors including the usage of only images from healthy subjects in the test dataset may also contribute to the high accuracy results. In fact, the MAE and RMSE of age estimation were found slightly smaller in the female group, and this reflects that any bias in data collection could have an impact on the model performance. Therefore, additional representative image data with reasonable variation for training could be considered in the future to minimize such biases. Furthermore, when our pretrained model is transferred to make inference in unseen datasets, locally adapted prediction models with the optimal predictive ability should be derived with the incorporation of local clinical images of the target population into the training set for better generalization. As exemplified in Figure 2A–D, the prediction model estimates the exact age, which is difficult for people to estimate from the images using the unaided eye. In this regard, this current study provides the first performance results of how deep learning is capable of extracting age information from CXR images and producing a prediction model using real-world-based data of healthy adults from the same hospital.

For age prediction, we demonstrated a series of SAMs for people of different age groups (Figure 8). We used Grad-CAM with regression method for visualization and averaged the activation by mapping it to a lung field template to minimize misregistration error, which we believe is more representative than to demonstrate only individual heatmaps. Some studies have reported age estimation from CXR images [39]. Hirotaka reported that the top of the mediastinum helps in predicting patients’ age [14]. Their results were similar to our study. However, they did not evaluate differences in heatmaps between different age groups. Our study showed that the activated degree and area differed across age groups. The first rib, cervical spine, lower thoracic spine, rib cage, and manubrium were the hot zones in the chest radiographs for those in their 20s, and the weighting of these regions decreased as people aged. The chondral ossification pattern of the skeleton varied according to time in a young life. Previous research mentioned that costochondral ossification of the first rib might start in the second decade and terminate at the third decade [40]. Its morphology is widely used in predicting the age of death in forensic fields [39,41]. Reports have mentioned that different ages have different thoracic kyphosis degrees between adolescence and adulthood [42], and degenerative change in spines is obvious in elderly persons. On the other hand, soft tissue of the thoracic wall and flank, although not a major contributor, continued to play a substantial role in the model across age groups. For radiographs of those older than 40, the group average of activation appears at the aortic arch and heart. Aortic arch diameter and the prevalence of aortic arch calcification increase with age. Aortic arch width progressively increases after four decades [43]. Therefore, it contributed more in patients aged over 40 years old. Considerable differences in cardiac size depending on sex and age are reported. The size changes significantly with increasing age [44]. Our study also showed that the heart region accounts more in older age groups in SAMs. The incidence of metabolic disorders such as hypertension or hyperlipidemia is higher in the elderly. Aortic arches begin to change first, followed by the hypertrophic change of the myocardium or heart chambers; our heatmap reflected the stages of physiological changes during aging. The incremental activation patterns across different age groups are consistent with the anatomical evolution in the thoracic region during the aging process, which proves the reliability of our model. In the current study, we also developed an accurate sex prediction model (AUC = 0.9999943). The presence of breast shadows and density, as well as the lower cervical spine, are the key activated locations of the sex prediction model. These regions of interest are consistent with those in previous studies [39,45]. Both MAE and RMSE were slightly smaller in the female group than male. The presentation of CXR-age, as a result of different aging and pathological processes, is expected to vary in different groups, also contributed to by different sexes. In summary, by using a deep learning approach, the imaging features can be quantified, and the model can jointly evaluate multiple anatomical systems, which thus enable more accurate age and sex predictions.

The model can be particularly useful when a person’s name and age/sex are unknown or unreliable, for instance, when a police officer or healthcare provider tries to help a person who has lost their memory or when a patient needs rapid imaging services at an emergency department with their name or age being unknown. Additionally, the model can be used in various forensic fields for subjects of unknown ages and sexes [2]. Chest radiographs are the most commonly used radiological test; therefore, they may be of potential use for security checks in places such as airports where identification of a fake ID is of critical importance.

The other significance is that age information in images is derived not only from chronological change, but also from the cumulative effects of biological and/or pathological change. Although our model could predict very accurately overall, the differences between predicted age and chronological age differed across individuals, which may contain information from changes from environmental factors, lifestyle, or diseases overtime for this individual. In fact, image-based age predictions have frequently been used to evaluate clinical conditions. For instance, bone age studies from hand and wrist radiographs are commonly used to evaluate the degree of skeletal maturity and diagnose growth disorders or to predict final adult height in children. In various brain diseases, neuroimaging-derived age predictions have been studied and the differences between the predicted brain and chronological ages are potentially related to the accumulation of age-related changes in pathological conditions [6,7,8,9,10,11] or protective influences on brain aging [46,47]. Similarly, CXR-age derived from chest radiographs may be of potential use as an imaging biomarker to represent the condition of the thorax. In fact, there have been emerging studies using CXR-age to successfully predict longevity, mortality, and cardiovascular risk [13,14,27], which provides a sound base for the imaging biomarker hypothesis.

In conclusion, the present study developed a deep learning model which can estimate age and sex in chest radiographs with a high accuracy than that in other radiographic methods. Future research to compare the accuracy of age and sex prediction models based on other imaging modalities and to further explore the relevance of CXR- age as an imaging biomarker is warranted.

## 5. Limitations

Our study has several limitations. First, we used a dataset comprising healthy Asian individuals from a single institution, which might introduce some potential biases in data collection. Therefore, the generalizability of our study should be addressed with additional studies on populations of different ethnicities, geographical regions, and socioeconomic levels. Additional representative image data with reasonable variation for training should also be considered to minimize such biases. Moreover, the deep learning model tendentiously underestimated age in individuals aged >60 years, potentially because a relatively low number of individuals from this age group were in the training set. Furthermore, the age-related features learned from the dataset might not be expressed as much as expected in older ages. Further investigation on a more balanced dataset may aid in clarifying this issue. Finally, although our model had a 2.1-year MAE and a 2.8-year RMSE, a more accurate and precise estimate for an individual is needed in the clinical settings. Further research with optimization of data manipulation or deep learning algorithms to reduce the error rate and to include different datasets to explore the relevance of CXR-age as a clinical imaging biomarker is warranted.

## Figures and Tables

**Figure 1 jcm-10-04431-f001:**
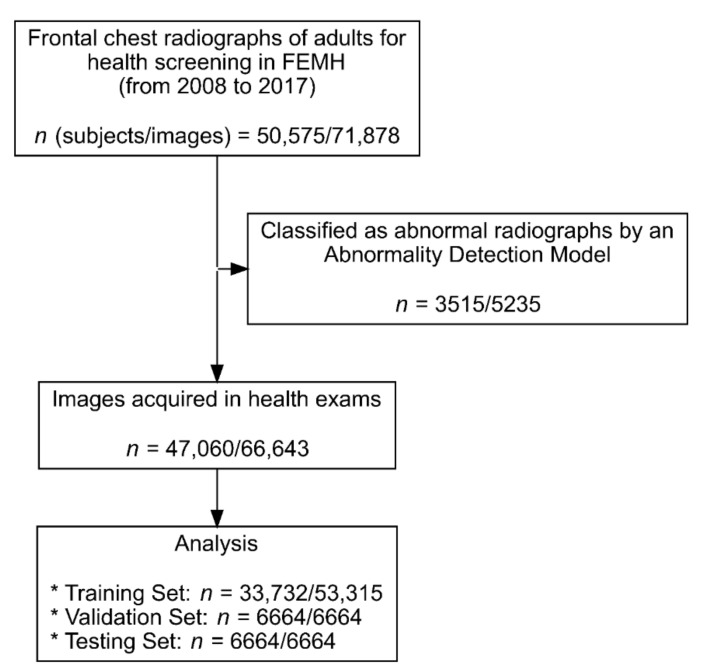
Recruitment and analysis flowchart.

**Figure 2 jcm-10-04431-f002:**
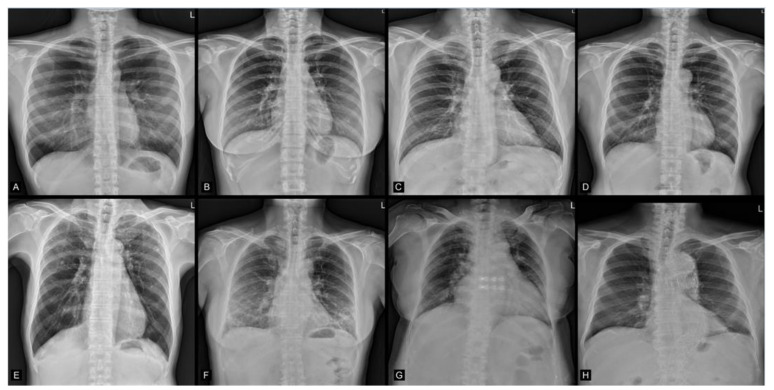
(**A**–**D**) Examples from the enrolled images. Chronological age/AI predicted age, sex/predicted sex: (A: 25/24.32, M/M), (B: 34/33.06, F/F), (C: 43/43.97, M/M), (D: 57/55.19, M/M) years old. (M = male, F = female). (**E**–**H**) Examples for the excluded images due to the presence of lung nodules, increased infiltration, cardiomegaly, or aortic stent-graft implantation, respectively.

**Figure 3 jcm-10-04431-f003:**
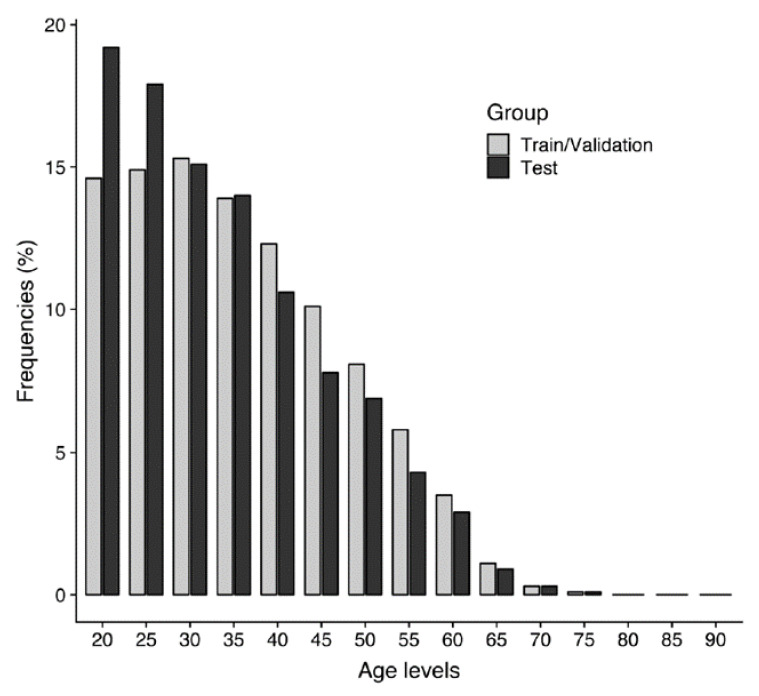
Distribution of chronological ages of the training and validation sets (*n* = 59,979) and testing set (*n* = 6664).

**Figure 4 jcm-10-04431-f004:**
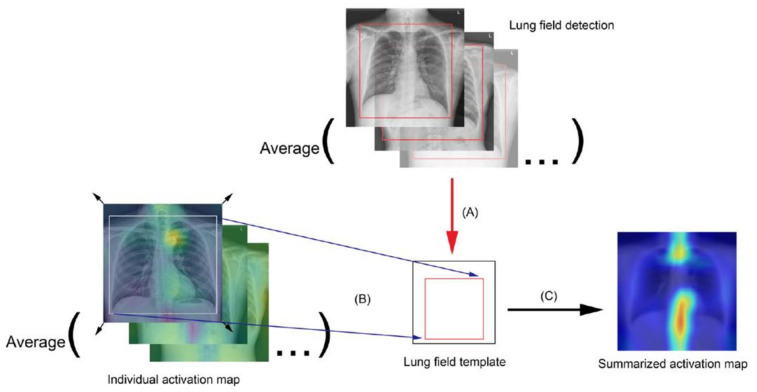
Flowchart for the summarized activation map (SAM): (**A**) Averaging the lung field bounding box coordinates, which were derived from an in-house build model, to generate a lung field template. (**B**) Transforming and mapping each individual activation map to the lung field template. (**C**) Averaging all the transformed activation maps to obtain the presented SAM, by age group.

**Figure 5 jcm-10-04431-f005:**
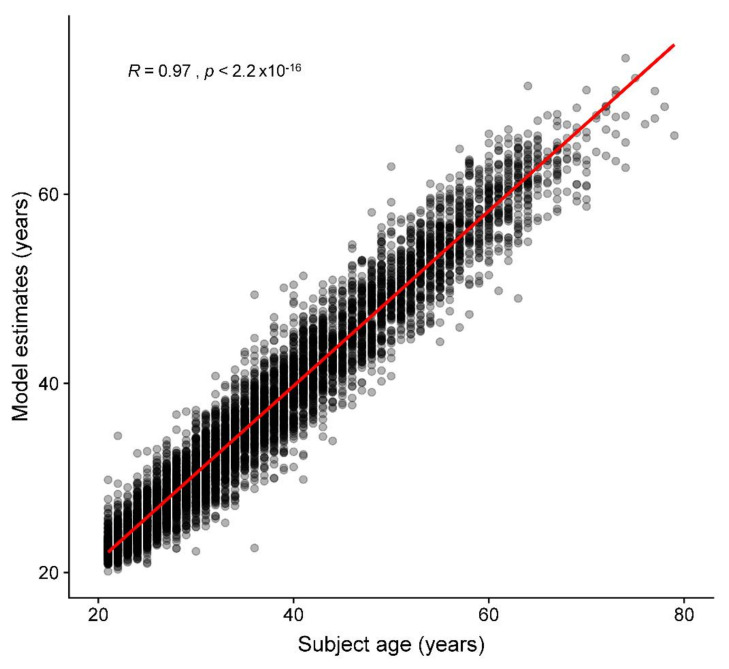
Accuracy of the deep learning model for age prediction. The scatter plot displays chronological age versus predicted age in the test set. The solid red line is the regression line. The R is the Pearson’s correlation coefficient of the model estimates with chronological ages.

**Figure 6 jcm-10-04431-f006:**
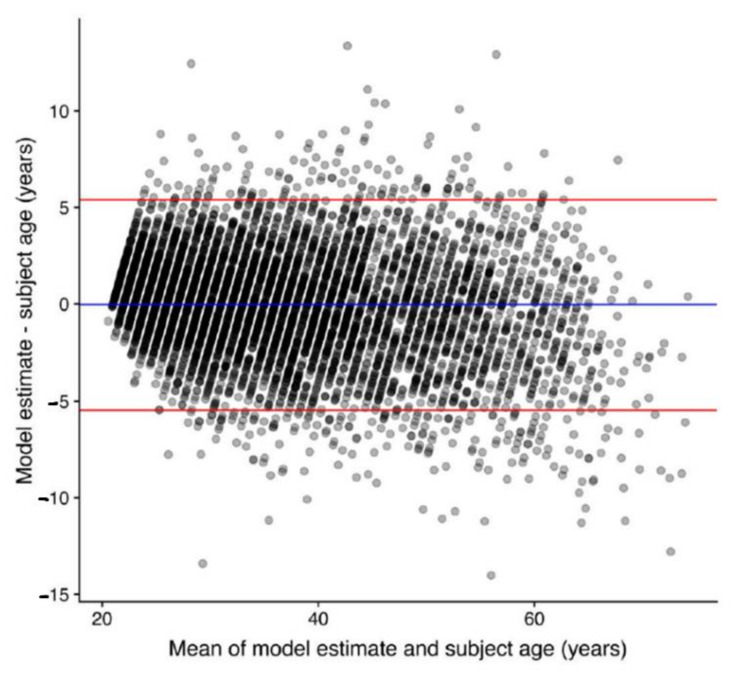
Bland-Altman plot, depicting the difference versus the mean between model estimates and chronological ages.

**Figure 7 jcm-10-04431-f007:**
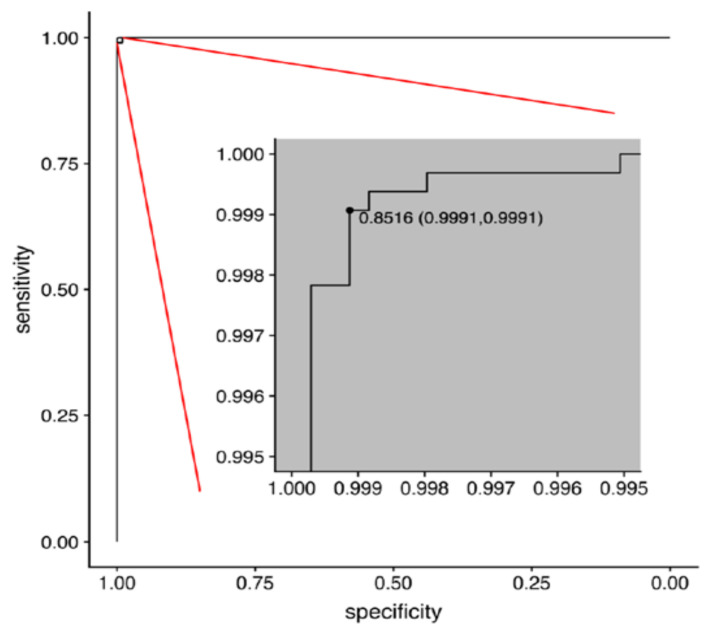
The receiver-operating characteristic (ROC) curve for the best trained deep learning model in sex prediction. The optimal threshold was selected by minimizing the Euclidean distance between the ROC curve and the (0, 1) point, which has a minimum value of (1 − sensitivity)^2^ + (1 − specificity)^2^. The optimal threshold of 0.8516 was also plotted on the ROC curve as the threshold (specificity, sensitivity).

**Figure 8 jcm-10-04431-f008:**
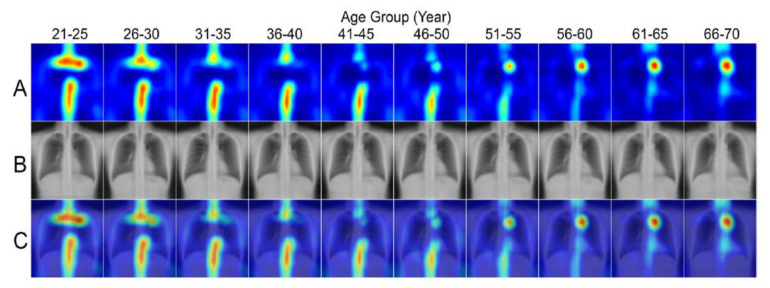
Summarized activation map (SAM) by age group. (**A**) Activation maps (**B**), the corresponding chest radiograph templates, and (**C**) the overlays across different age groups.

**Table 1 jcm-10-04431-t001:** Image-based demographics of individuals of the training, validation, and test datasets.

Set	Training (*n* = 53,315)	Validation (*n* = 6664)	Test (*n* = 6664)	Total (*n* = 66,643)
Age				
Mean (Standard deviation)	39.1 (11.9)	37.1 (11.8)	36.8 (11.8)	38.7 (11.9)
Median (Min, Max)	38.0 (20.0, 93.0)	35.0 (20.0, 86.0)	35.0 (20.0, 79.0)	37.0 (20.0, 93.0)
Sex				
Female	29,104 (54.6%)	3490 (52.4%)	3440 (51.6%)	36,034 (54.1%)
Male	24,211 (45.4%)	3174 (47.6%)	3224 (48.4%)	30,609 (45.9%)

## Data Availability

The data presented in this study are available on request from the corresponding author. The data are not publicly available due to the restriction of IRB.

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
