# Peer review of "Using Deep Neural Networks for Predicting Age and Sex in Healthy Adult Chest Radiographs"

_jcm, 2021, doi:10.3390/jcm10194431_

Round 1

Reviewer 1 Report

This manuscript proposes a deep learning model for prediction of age and sex using chest radiographs. The application is novel. However, I think that there are several concerns that must be be addressed first.

1. The organization of the manuscript should be improved. There are several places where the content is presented in a non-standard way. For example, the last few sentences of the introduction describe study design and the first section of results describe the data. These are just examples and there are many other places where such inconsistencies are present. The authors should carefully revise the manuscript.

2. The results are difficult to believe/understand. The authors make no effort to explain why the method works and whether there can be any biases that might make the results overly optimistic. One of the most important source of bias for both age and sex prediction can arise due to body size/weight. What is the authors take on this? If this information is available then the authors should provide evidence that the models are not biased. If such information is not available then can the authors provide any alternative ways to address this? One possibility is to make use of the fact that usually males have larger bodies than females, so in this case are there any biases/differences in age prediction for males and females? Given the potential clinical application as proposed by the authors and the journal, such considerations must be addressed.

3. The authors make no attempt to discuss whether the activation maps are meaningful. Also it is unclear why the sex prediction activation maps are not shown.

4. Can the authors make their code and models available for inspection?

Author Response

Reviewer #1:

  1. The organization of the manuscript should be improved. There are several places where the content is presented in a non-standard way. For example, the last few sentences of the introduction describe study design and the first section of results describe the data. These are just examples and there are many other places where such inconsistencies are present. The authors should carefully revise the manuscript.

Response:

  • Thank you for the helpful comments. We have revised the manuscript accordingly, trying to make it more organized and more readable.
  • We have moved the “3.1 Study population demographics in Result section” to the Materials and Methods 2.3 (line 124-129, page 4).
  • “The last few sentences of the introduction which describe study design” was deleted because of duplication with Materials and Methods 2.5 and 2.6 (line 145-195, page 5-6).

  1. The results are difficult to believe/understand. The authors make no effort to explain why the method works and whether there can be any biases that might make the results overly optimistic. One of the most important source of bias for both age and sex prediction can arise due to body size/weight. What is the authors take on this? If this information is available then the authors should provide evidence that the models are not biased. If such information is not available then can the authors provide any alternative ways to address this? One possibility is to make use of the fact that usually males have larger bodies than females, so in this case are there any biases/differences in age prediction for males and females? Given the potential clinical application as proposed by the authors and the journal, such considerations must be addressed.

Response:

  • Thanks for this suggestion. To answer reviewer’s question about the differences of CXR-age estimation in different sexes, we conducted t test for absolute errors and squared errors between groups and found that there was only a slight difference in MAE (F: 2.1, M: 2.2, p=0.08) and in RMSE (F: 2.7, M: 2.8, p=0.04). This additional result is included in the revised Result 3.1 (line 228-230, page 8). We have also discussed the chest age, as a result of different aging and pathological processes, which is expected to vary across sex and different age groups in the revised Discussion (line 331-335, page 11).
  • We agree that there might be some bias in any model training process. We think that different variations, including body size/weight, in each group, can provide information and have contributed to the estimation of the CXR-age. Theoretically, the prediction of age/sex will be more accurate if combining more signs or clues (e.g., body size/cartilage ossification, soft tissue shadow of thoracic wall, heart size or size of aortic arch). However, detection of small differences between multiple signs is the limitation of unaided human eyes. Deep learning technique, by contrast, has potential for detecting these corporate changes in a quantitative way. We have discussed the related issues in the revised manuscript. Please refer to Introduction (line 56-63, page 2) and Discussion (line 297-335, page11).
  • Although, we don’t have data of body size or weight but we do believe that with a sufficient data size the image presentation of body size could be well learned and adjusted by the deep learning model.
  • We think the main reason for the higher accuracy of our model is the homogenous of our dataset. To develop a model that minimizes non-physiological change of aging, we used only high-quality chest radiographs and excluded images from people with lung diseases, trauma, surgical condition, or any type of implants. Please refer to Discussion (line280-296, page 10-11).
  • We have also added the potential clinical application of our prediction method in the Discussion (line 336-360, page 11-12).

  1. The authors make no attempt to discuss whether the activation maps are meaningful. Also it is unclear why the sex prediction activation maps are not shown.

Response:

  • Thank you for reviewer’s suggestion. We have reviewed the age-related image signs for correlation of our activation heatmap. We have revised the paragraph and discussed our results of activation map providing literature and more detailed results accordingly. Please refer to the Discussion (line 297-335, page 11).
  • With regards to sex prediction map, it is straightforward that the map showed model using breast shadow to predict sex. Therefore, we do not include the activation map of sex in the revised manuscript. Instead, we described the related results in Result 3.3 (line 252-253, page 9) and Discussion (line 327-331, page 11).

  1. Can the authors make their code and models available for inspection?

Response:

  • Thank you for reviewer’s question. It would be difficult to share the code or model at this stage because this is related to a collaborative project with a commercial company. There are certain restrictions in sharing codes or models. However, as we only modified the activation function to sigmoid function for age prediction model and still keep softmax function for sex prediction model. Therefore, it would be easy to modify from the original inception-resnet-v2 code. Please refer to this address for model’s code. We also listed it in the Material and Method 2.4. (line 133-137, page 5) (https://github.com/keras-team/keras/blob/master/keras/applications/inception_resnet_v2.py)

Reviewer 2 Report

What is the significance of the proposed work, as sex and age can be detected in a non-invasive manner?

What is the significance of chest radiographic images in the detection of sex and age when the same can be detected through face images?

There are no technical details of the model. All the concepts are given only in an abstract manner which is not acceptable at this level. 

A review of existing work in the same domain is not discussed.

Results are not presented properly, must compare the results with some existing work found in the literature.

The conclusion section is very weak, it must be rewritten.

Author Response

Reviewer2

  1. What is the significance of the proposed work, as sex and age can be detected in a non-invasive manner?

Response:

  • Thank you for reviewer’s comment. We are sorry that we did not make it clear in the original manuscript. We have added the potential clinical application of our prediction model in the revised manuscript. Please refer to Discussion (line 336-360, page11-12).

  1. What is the significance of chest radiographic images in the detection of sex and age when the same can be detected through face images?

Response:

  • Thank you for reviewer’s comment. There are two reasons that CXR predicting age could be better than face image predicting age.
  • First, there is opportunity that human could tell how old of a person through face image perception, but the deep learning technique can corporate subtle changes in a quantitative way and undoubtedly can provide greater accuracy than unaided human eyes. We have described this in the Introduction (line 49-63, page 2) and in the Discussion (line 331-335, page 11).
  • Second, although chronological age has been widely used in various studies to predict disease and treatment prognosis, biological age may better reflect the impacts of lifestyle, nutrition, multiple risk factors, and environmental factors, which, in turn, can benefit the prediction of disease and prognosis both theoretically and practically. Among all medical imaging modalities, chest radiograph is the most widely available modality, displaying plenty of information about the cardiopulmonary and musculoskeletal system. Using chest-radiograph-derived biological age estimates to successfully predict longevity, long-term mortality and cardiovascular risk were also reported in the literature, but none of them have validated the prediction model in a real-world clinical dataset. Thus, we aimed in this study to assess the accuracy of a deep learning model for age and sex estimation based on chest radiographs of a healthy adult cohort from a real-world clinical dataset. We have added the related explanations in Introduction (line 77-91, page 2) and Discussion (line 344-360, page 11-12).

  1. There are no technical details of the model. All the concepts are given only in an abstract manner which is not acceptable at this level.

Response:

  • Thank you for reviewer’s comment. We have rewritten the methods concerning of building the model. The CNN architecture is addressed in Material and Method, 2.4 (line 133-144, page 5).
  • The detailed pre-processing and training processes are shown in Material and Method, 2.5 (line 145-173, page 5-6).
  • The post processing (group summarized activation map) is shown in Material and Method, 2.6 (line 174-195, page 6).

  1. A review of existing work in the same domain is not discussed. Results are not presented properly, must compare the results with some existing work found in the literature. The conclusion section is very weak, it must be rewritten.

Response:

  • Thank you for reviewer’s suggestions. We have reviewed the existing literature, including other imaging modalities predicting age, and chest radiographs predicting age, and added them in the revised manuscript. Please refer to introduction (line 38-48, page 1) and discussion (line 280-296, page 10-11).
  • We reviewed the age-related image signs for correlation of our activation heatmap. We re-wrote the paragraph and compared our results of activation map with the existing literatures in more details accordingly. Please refer to the discussion (line 297-335, page 11).
  • We also reviewed the existing literatures of potential use of clinical applications for chest age in Introduction (line 49-59, page 2), and Discussion, (line 336-360, page 11-12).
  • We have revised the conclusion according to reviewer’s suggestion. Please refer to the line 361-365, page 12).

Round 2

Reviewer 1 Report

The authors has reorganized the manuscript and now it follows a more standard format.

However, I think that my concerns regarding potential biases in the data which in turn can be learned by the CNN are misunderstood by the authors. My concern is that the biases, e.g. due to males having a higher body size, can bias the models and in the border cases the predictions will go wrong. The authors now try to spin this as a positive thing (learning from additional clues) but this is exactly the wrong thing to do, especially for corner cases and the aim should be to learn models that are independent of such biases. It is understandable that the authors do not have the required data to actually test/control such biases but the authors should discuss this in detail and acknowledge limitation(s) instead of trying to turn this into a positive point.

I also do not completely agree that the high accuracy is simply because of the high quality data. Unlike the biases this point is actually easy to check. Can the authors train their models after including the "bad" data and show its impact on the accuracy?

Author Response

  1. The authors have reorganized the manuscript and now it follows a more standard format. However, I think that my concerns regarding potential biases in the data which in turn can be learned by the CNN are misunderstood by the authors. My concern is that the biases, e.g. due to males having a higher body size, can bias the models and in the border cases the predictions will go wrong. The authors now try to spin this as a positive thing (learning from additional clues) but this is exactly the wrong thing to do, especially for corner cases and the aim should be to learn models that are independent of such biases. It is understandable that the authors do not have the required data to actually test/control such biases but the authors should discuss this in detail and acknowledge limitation(s) instead of trying to turn this into a positive point.

Response

We are sorry that we misunderstood the meaning. The MAE and RMSE of age estimation were found slightly smaller in the female group. This reflects that any bias in data collection could have an impact on the model performance. We have revised the manuscript accordingly, discussing the potential bias in the discussion (line 292-300, page 11), and added more descriptions in the limitation. (line 375-380, page 12)  

  1. I also do not completely agree that the high accuracy is simply because of the high quality data. Unlike the biases this point is actually easy to check. Can the authors train their models after including the "bad" data and show its impact on the accuracy?

Response

Thank you for this valuable opinion. We completely agree with the reviewer that the high accuracy is not simply because of the usage of high-quality images. Other factors such as the usage of only images from healthy subjects in the test dataset or any other potential bias may also contribute to the high accuracy results. The MAE/RMSE results cannot be directly compared across different studies either. We have revised the manuscript accordingly (line 288-291, page 11 and line 292-300, page 12). We also agree that including the “bad” data or additional image data with variation for training could provide more information about model performance and minimize the biases. However, we are unable to perform this analysis because this study was only approved by the IRB as simply research based on a clinical dataset from a healthy adult cohort from a single institution. Further research with optimization of data manipulation or deep learning algorithms and to include different datasets to explore the relevance of CXR-age as a clinical imaging biomarker is warranted. We have revised the manuscript accordingly (line 375-380, page 12, and line 387-389, page 12).

Reviewer 2 Report

kindly check the manuscript English through a professional tool like the premium account of Grammarly.

Author Response

Thank you for your opinion. We have followed the suggestion, checking the revised manuscript with Grammarly and re-edited it using the journal English editing service.

Round 3

Reviewer 1 Report

The authors have adequately addressed most of my concerns. The only remaining concern is that the response regarding IRB limitations does not seems to make sense and this is likely due to misunderstanding. As the authors did quality control of the data and rejected some images of bad quality they can easily reintroduce these rejected "bad" images in their training data. I do not see how this can conflict with the existing IRB approvals as the authors already have access to this data.
